# Cognitive correlates of math abilities in autism spectrum disorder

**Irene Tonizzi** *, **M. Carmen Usai**

Department of Educational Sciences, University of Genoa, Genoa, Italy

* irene.tonizzi@edu.unige.it

## Abstract

The purpose of the current study was to investigate the contribution of different cognitive processes to specific math abilities in students with autism spectrum disorder (ASD) and typically developing (TD) students. The study involved a group of students with ASD without intellectual disabilities (n = 26) and a group with TD students (n = 52). The two groups aged from six to 20 years old and were matched for age, sex ratio and visuospatial reasoning. To assess math abilities, four math tasks were administered: arithmetic facts, mental calculation, mathematical inferences and math problem solving. Concerning cognitive processes, participants were tested on vocabulary, verbal working memory, visuospatial working memory, response inhibition and interference control. The group with ASD showed lower scores on all specific math measures than the TD group; cognitive processes differently contributed to diverse math abilities, and vocabulary and verbal working memory were stronger associated to specific math abilities in the group with ASD than in the TD group. The current results suggest that students with ASD had lower math abilities that are generalized to different math tasks. Implications for research and clinical assessment and intervention were discussed.

**Data Availability Statement:** All relevant data are within the manuscript and its Supporting Information files.

**Funding:** The author(s) received no specific funding for this work.

## Introduction

Previous literature suggested that children with ASD without intellectual impairment may show relative weaknesses in certain subdomains of mathematics [1, 2]. Deficits in inhibitory control (IC) and working memory (WM), found in previous studies and meta-analyses [3–5] may account for the discrepancy among students with ASD that show average or above-average level of intellectual functioning but difficulties in some mathematics areas. In the literature on typical developing, recent studies suggested that the contribution of IC and WM may differ across different components of mathematical learning [6–8]. However, the relationship between IC, WM and specific math competencies remained quite unexplored in children with ASD [9].

Understanding the relative points of strength and weakness in mathematical learning in children with ASD and investigating to what extent IC and WM contributed to these abilities may be of fundamental importance to deepen our knowledge of learning processes in ASD and improve educational strategies for these children.

**Competing interests:** The authors have declared that no competing interests exist.

## Math abilities in autism spectrum disorder

Autism spectrum disorder (ASD) is a neurodevelopmental condition characterized by challenges in social communication and the presence of restricted or repetitive behaviors, interests, or activities. These characteristics are seen across diverse cultural, racial, ethnic, and socioeconomic backgrounds status [10]. The last edition of both diagnostic manuals, DSM-5 and ICD-11, highlighted the high heterogeneity of ASD, which arises from different levels of symptoms, intellectual and linguistic abilities, and the presence of comorbidities [11]. They also distinguished between ASD with or without intellectual disabilities and functional language impairment, with ICD-11 that even provided different diagnostic codes. In addition, in ICD-11, it is acknowledged that some individuals with ASD may not show evident social challenges and distress in childhood, as their ability to adapt to various contexts requires exceptional effort; thus, the core features of ASD may only become fully apparent in adolescence or adulthood when social demands surpass their capacities. The new approach proposed by ICD-11 highlighted the importance of better understanding the features associated with autism, even in children, adolescents, and adults who do not have cognitive disabilities, but whose difficulties imply a significant effort to adapt to social contexts [12]. Concerning academic achievement, it is worth to note that a significant group of students with ASD without concurrent intellectual disability still struggle to achieve their full potential in educational settings without proper support [1].

Understanding their academic strengths and weaknesses is vital, especially since many are in regular education settings, and this is even true for STEM disciplines that have a high impact on academic and professional career [13, 14]. However, research on math achievement in individuals with autism is limited; this gap may be due to the idea supported by the "male brain theory" that people with autism have a preference for rule-based fields and have exceptional math abilities; however, evidence for this is mostly anecdotal [15, 16].

In fact, only a small proportion of individuals with ASD display exceptional mathematical talents and mathematical challenges appear to be more prevalent among students with ASD compared to their typically developing (TD) peers [17, 18]. However, research on math achievement in ASD is limited and often yields inconsistent results [1]. Some studies suggest better math abilities in students with ASD than in their TD peers [19, 20], while others show the opposite outcomes [21, 22]. A previous meta-analysis showed a significant small-to-medium difference between the two groups in the performance of math tasks [23], in contrast with the stereotype of increased mathematical proficiency among individuals with ASD, which has been supported by some descriptive studies [15, 16].

In terms of strengths and weaknesses within the specific domain of mathematics, a part of literature suggested that children with ASD tend to excel in fact-based math, such as recalling arithmetic facts, but struggle with more complex math tasks, including solving word problems and equations [1, 2, 24, 25]. Recent studies also indicate that students with ASD perform worse in problem-solving tasks compared to computation tasks, which is not seen in typically developing or ADHD groups [22, 26]. However, a previous meta-analysis found that the difference between participants with ASD and TD did not vary with the type of math task, suggesting that students with ASD show similar performance on different math tasks [23]. However, it's important to note that this result was based on an analysis of the two most commonly used tasks, which are numerical operations and problem solving, and did not allow for an exploration of other math domains that may involve different processes and knowledge; for example, a still underexplored domain is represented by arithmetic facts, that require a factual mathematical knowledge that can be more easily automatized, compared to mathematical conceptual knowledge implied, for example, in mathematical reasoning tasks [6].

Also, domain-general cognitive processes play a significant role in explaining differences in math performance between students with ASD and TD peers. While intellectual functioning has consistently been linked to math achievement in TD students [e.g., 27] the relationship between intelligence and math performance in students with ASD is less clear. The aforementioned meta-analysis suggested that Verbal IQ plays a significant moderating role in the relationship between IQ and math performance in participants with ASD [23]. This finding aligns with earlier research indicating that verbal subtests on tests like the WISC-III and WISC-IV have stronger associations with academic achievement than performance subtests [28]. Additionally, research has suggested that early atypical language development in ASD may lead to the use of less efficient strategies in verbal problem-solving tasks [22, 29].

Notably, recent studies and reviews have also proposed that, in addition to intellectual functioning, impairments in working memory (WM) and inhibitory control (IC) may affect mathematics achievement in students with ASD.

## Working memory, inhibitory control and math abilities in autism spectrum disorder

According to recent meta-analyses, children with ASD may show significant impairments in both WM [5, 30] and IC [4, 31]. Working memory is the ability to actively elaborate the material in mind by adding new information and, for example, allows children to keep in mind and update data in mental calculations [6]. IC is the ability to suppress automatic responses and ignore distracting stimuli to perform alternative behaviors; it is necessary, for example, in the selection of data when irrelevant information is present in the text of a problem or in suppressing an overlearned strategy in favor of a less dominant response [32]. The relative strengths and weaknesses in certain mathematics skills, in conjunction with their difficulties in WM and IC suggested specific connections between these cognitive processes and mathematics in children with ASD. However, despite the extensive literature on WM and math skills in TD children [6, 33, 34], relatively little is known about this relationship in autism [20, 22, 35–37]. Some studies have suggested that WM impairments, especially in the verbal and central components, plays a fundamental role in predicting mathematics performance in ASD, both in computation and problem-solving tasks [22, 38]. Interestingly, Wang and coauthors [39] suggested that WM impairment in preschool may represent the main cause of later math difficulties in autism, suggesting that strong early WM may help children with ASD catch up with their peers in math. Differently, a measure of response inhibition (i.e., Day/Night Stroop) was not a significant predictor of early math abilities in this sample of preschoolers. Also, in a study conducted by Polo-Blanco and co-authors [40] with older children, inhibition was not a significant predictor of math problem solving when the entire sample of participants with ASD was considered. However, poorer performers (i.e., children who obtained ≤ 25% correct answers) showed lower scores in inhibition, theory of mind and verbal comprehension. Interestingly, the authors also found a connection between the degree of abstraction employed in their strategies during the resolution of math problem and three cognitive factors: inhibition, cognitive flexibility, and theory of mind. Surprisingly, this correlation was found only in the sample of children with ASD and absent among the non-ASD group. Notably, both studies used a response inhibition measure, while to our knowledge, no previous study investigated the effect of interference control on math abilities in participants with ASD, although previous research have suggested that the ability to filter out distractors may be more related to mathematical learning [8, 32]. In addition, studies addressed the role of WM and IC adopted math problem solving [40] and numerical operation [22] without considering other math abilities.

Thus, a detailed investigation of the contribution of specific cognitive processes to different math knowledge in students with ASD is needed.

## Theoretical framework

Studies focusing on typical development suggested that the contribution of working memory and inhibition may differ across specific components of math ability [6–8]. To investigate in depth the math profile and the cognitive underpinning of math abilities in students with ASD, as compared with TD participants, it is necessary to consider both the multidimensionality of math abilities and the specific contribution of domain-general processes. In this regard, the model proposed by Cragg et al. [6] for identifying the cognitive correlates of math abilities in typical development could be helpful to better understand these associations also in autism. In fact, the authors [6] considered math abilities as a multidimensional construct that involve three types of math knowledge: factual knowledge (i.e., the ability to retrieve arithmetic facts from long term memory), procedural skills (i.e., the ability to both know which procedure to follow and complete the appropriate steps to arrive at the correct answer, for example in resolving arithmetic word problems or mental calculations) and conceptual knowledge (i.e., ability to understand the mathematical principles or relationships that underlie the targeted concept, for example the understanding of the properties of the operations). The model indicated that WM contributed to all math knowledges, whereas IC was associated with factual knowledge and procedural skill [6]. Based on recent literature [17, 24], it could be hypothesized that students with ASD may encounter more difficulties in procedural and conceptual understanding, than in factual knowledge. In addition, math tasks that require complex procedural skills and the understanding of abstract math concept could be more challenging for students with ASD and they may rely more on IC and WM. According to previous studies, verbal working memory may play an important role in math abilities in both participants with ASD and TD [22, 23]. However, it is also possible that visuospatial WM could explain a portion of the difference between the ASD and TD groups in math achievement; for example, visuospatial WM could play an important role in decomposition strategies in solving numerical operations but also in generating the mental representation of math word problems [6]. For this reason, studies assessing the contribution of visuospatial WM in participants with ASD are needed. The contribution of IC to math abilities was less clear in TD students [6, 41] and rarely addressed in studies with participants with ASD; in addition, these studies adopted only response inhibition tasks and not interference control tasks [39, 40]. In addition to WM and IC, also verbal intellectual abilities may impact on specific math abilities, especially in participants with ASD, in line with the hypothesis that WM and verbal IQ may account for a significant portion of variability in math performance in students with ASD [22, 38, 42].

In summary, there is a need to investigate the mechanisms by which cognitive processes supports math achievement in students with ASD, as compared with TD students, adopting a multidimensional approach that investigate the specific contribution of cognitive processes to different math abilities.

## The present study

The first aim of the current study was to investigate whether the group with ASD shows poorer math abilities than the TD group. As reported above, studies on this issue often showed heterogeneous results [1, 17]. with some studies that found poorer math abilities in autism [22, 43] others reporting no differences [44], and still others finding better math abilities in participants with ASD [19, 20]. A previous meta-analysis [23]. showed a statistically significant difference with a small-to-medium effect size between the two groups, with a poorer math

performance in students with ASD. However, previous studies generally used one or two math tasks (mainly numerical operations and math problem solving) and, generally did not include other types of math skills, such as arithmetic facts [e.g., 22, 36]. As some studies suggested that students with ASD could encounter more difficulties in more abstract and complex math tasks that mainly required conceptual knowledge [24, 25] we investigated math abilities in group with ASD and in a TD group, using different tasks intended to measure specific types of math knowledge. Therefore, we adopted a battery of math tasks that included arithmetic facts, mental calculation, mathematical inferences and math problem solving tasks. It is conceivable that the participants with ASD showed more difficulties in the last two tasks (mathematical inference and math problem solving) as they are more complex tasks, requiring more conceptual knowledge than the other two tasks; in fact, arithmetic facts mainly measured factual knowledge and mental calculations mainly measured procedural math knowledge, together with factual knowledge [6].

The second aim of the study was to examine the contribution of domain-general cognitive processes to math abilities. Therefore, we investigated which cognitive processes among vocabulary, verbal and visuospatial WM, response inhibition and interference control contributed to specific math abilities, and if this association differed between the two groups. The previous meta-analysis highlighted the role of verbal working memory on explaining group's differences between the two group, but the role of visuospatial working memory and inhibitory measures remained quite unexplored [20, 40]. In summary, the study aimed to answer to the following research questions:

a. Are there differences between the group with ASD and the TD group in specific math abilities? Are there relative areas of strength (e.g., arithmetic facts and mental calculations) and areas of weakness (e.g., mathematical inferences and math problem solving)?

b. Which cognitive processes contributed to specific math abilities, over and above vocabulary? Does this contribution differ between the group with ASD and the TD group?

## Method

### Participants

The study involved a group of participants with ASD and a comparison group of TD participants, composed of children, adolescents and young adults. Concerning the inclusion criteria, participants of both groups had to be over six years and scored at or above the average (Scaled Score ≥ 8) on the Matrix Reasoning subtest of the WISC-IV or the WAIS-IV. The choice to use a sample with an age older than six years is due the fact that, at this age, the two dimensions of IC should already be distinguished, although still under development [45]. Concerning the group with ASD, all participants had a previous formal diagnosis of ASD according to the criteria of the Diagnostic and Statistical Manual of Mental Disorders, 5th edition or 4th edition, Text Revision or International Classification of Diseases 10th edition or 11th edition. This formal diagnosis was confirmed by a score above the ASD cut-off (T-score > 60) on the Italian version of the Social Responsiveness Scale, Second edition (SRS-2 [46]) and a score above the ASD cut-off on the Childhood Autism Rating Scale, Second Edition (CARS-2 [47], ASD cut-off corresponds to a score above 28 in the CARS-2-High Functioning Version and above 30 in the CARS-2-Standard Version). Concerning the TD group, only participants without any previous diagnosis and with a T-score < 60 in the SRS-2 were included.

As for the ASD group, 33 participants joined the study; however, only 26 participants were included. In fact, seven of them were excluded because they did not meet the inclusion criteria (n = 3 excluded due to the lack of a formal diagnosis of ASD; n = 4 excluded due to a scaled

**Table 1. Age and cognitive measures of the group with ASD and the TD group.**

| | ASD group | | | | | TD group | | | | | | |
|---|---|---|---|---|---|---|---|---|---|---|---|---|
| | N | M | SD | Min | Max | N | M | SD | Min | Max | t test | Cohen's d |
| Age | 26 | 11.99 | 3.21 | 6.41 | 19.12 | 52 | 12.41 | 3.58 | 7.18 | 19.53 | t(76) = 0.50, p = .618 | 0.12 |
| Matrix Reasoning | 26 | 12.35 | 2.86 | 8.00 | 17.00 | 52 | 12.25 | 2.47 | 8.00 | 18.00 | t(76) = -0.15, p = .878 | -0.04 |
| Vocabulary | 26 | 10.23 | 3.34 | 2 | 15 | 52 | 11.85 | 2.65 | 6 | 19 | t(76) = 2.32, p = .023 | 0.56 |
| BDS | 26 | 6.88 | 2.14 | 3 | 13 | 52 | 7.75 | 1.86 | 4 | 12 | t(76) = 1.84, p = .069 | 0.44 |
| Mr. Peanut | 22 | 6.32 | 2.88 | 3.00 | 13.00 | 51 | 8.12 | 2.70 | 3.00 | 16.00 | t(71) = 2.56, p = .013 | 0.65 |
| MFFT errors | 25 | 8.28 | 6.93 | 0 | 24 | 50 | 4.86 | 3.96 | 0 | 18 | t(73) = -2.72, p = .008 | -0.67 |
| MFFT RT | 25 | 27.49 | 12.38 | 11.82 | 58.83 | 50 | 20.55 | 10.47 | 7.47 | 56.59 | t(70) = -2.47, p = .016 | -0.62 |
| Flanker AI | 22 | 0.90 | 0.10 | 0.60 | 1.00 | 48 | 0.96 | 0.05 | 0.77 | 1.00 | t(68) = 3.34, p = .001 | 0.86 |
| Flanker RTI | 22 | 1063.44 | 394.71 | 513.97 | 1813.17 | 48 | 943.07 | 333.03 | 427.50 | 1925.35 | t(68) = -1.32, p = .190 | -0.34 |

score on the Matrix Reasoning subtest ≤ 8). As for the control group, 62 participants joined the study, but we had to exclude from the statistical analysis 10 participants because five of them did not meet the inclusion criteria (n = 4 had a previous diagnosis of Learning Specific Disorder, n = 1 had a previous diagnosis of Attention-deficit/hyperactivity disorder) and the other five have only completed the first session.

Therefore, the final sample was composed of 78 participants (n = 26 for the group with ASD and n = 52 for the TD group). As shown in Table 1, the two groups were matched for age and visuo-perceptual reasoning measured with the Matrix Reasoning subtest; however, they significantly differed in the Vocabulary subtest (WISC-IV or WAIS-IV), where the TD group obtained higher score than the ASD group. Moreover, concerning other cognitive variables, the two groups significantly differ in visuospatial working memory (Mr. Peanut) but not in verbal working memory (Backward digit span, BDS); a significant difference between the two groups was also found in the response inhibition task (Matching Familiar Figure tasks, MFFT) in both errors and RTs, and in the accuracy in the interference control task (Flanker task).

## Procedure

The study began on 1st December 2020, and concluded on 8th January 2023.As the study started during the COVID-19 pandemic, the tasks were adapted to be administered online. This modality was used for the entire duration of the study to avoid possible effects due to administration mode. Each participant was tested individually during online video calls. Specifically, the tasks were administered in three online sessions: the first session included vocabulary, matrix reasoning and backward digit span; the second session included Matching Familiar Figures task, Mr. Peanut and Flanker task and the third session included four math tasks. All the tasks requiring visual materials were administered using screen sharing (Matrix Reasoning, Vocabulary, Matching Familiar Figures task). For Mr. Peanut and Flanker tasks a computerized version was used, administered through Inquisit Web. in these cases, the links were sent to participants, and they were asked to share their screen during the task. For participants with ASD, an additional in-person session was included to administer the CARS-2 that required the direct observation of participants' behavior. Parents of each participant were asked to complete the SRS-2.

## Measures

**Vocabulary and visuospatial reasoning measures.** Vocabulary subtest and Matrix Reasoning of the WISC-IV [48] were administered to participants between the ages of 6 and 16 years and the same subtests of the WAIS-IV [49] were administered to participants over 16 years. Scaled scores were used.

*Vocabulary (VC).* It is an important indicator of Verbal Comprehension Index and measures participants' verbal fluency and concept formation, word knowledge, and word usage. In this subtest, participants are asked to define a given word. Scaled score ranged from 1 to 19. The WISC-IV Vocabulary subtest [48] was administered to participants between the ages of six and 16 years and the WAIS-IV Vocabulary subtest [49] was administered to participants over 16 years. Scaled scores were used (range 1–19).

*Matrix Reasoning (MR).* It is an untimed core subtest of Perceptual Reasoning Index that measures visual processing and abstract, spatial perception. Participants are shown colored matrices or visual patterns with a missing piece. The participant is asked to select the missing piece from five alternatives. The WISC-IV Matrix Reasoning subtest [48] was administered to participants between the ages of six and 16 years and the WAIS-IV Matrix Reasoning [49] was administered to participants over 16 years. Scaled score were used (range 1–19).

**Working memory measures.** *Backward digit span (BDS; [48, 49]).* The Backward digit span task requires the participant to repeat numbers in reverse order. There is no time limit for the participant to respond, but the examiner reads each number out aloud at the rate of one number per second. The task is composed of eight levels with increasing difficulty; each level was composed of two items and the number of digits to remember increases by one every level from two to nine digits. One point was given for each correct trial (Backward digit span, expected range 0–16). The WISC-IV Backward Digit Span [48] was administered to participants between the ages of six and 16 years and the WAIS-IV Backward Digit Span [49] was administered to participants over 16 years. Row scores were used (range 0–16).

*Mr. Peanut.* This task is considered a measure of visuospatial WM [50, 51]. We used a computerized version of this task. Participants were shown a character, Mr. Peanut, with a number of coloured stickers attached to different parts of his body (e.g., on the right leg, on the nose, etc.) for five seconds. Then, Mr. Peanut disappeared and reappeared without stickers. The participants had to indicate the position and the colour of the stickers as they were presented in the previous figure. There are three items per level (from 1 to 7 stickers). An item is scored as correct if the participants select the correct coloured stickers and locate them in the correct body parts. If a level is successfully mastered (at least one correct attempt per level), participants move up a level. If all 3 attempts per level fail, the test concludes. Test-retest reliability (Pearson's r) calculated in 75 TD children was .39, p < 0.001) [52]. The total of correct items was registered (Mr. Peanut, expected range 0–21).

**Inhibitory control measures.** *Matching Familiar Figure task (MFFT; adapted from [53]).* This task is considered a measure of response inhibition because the participant is required to control the tendency to respond before evaluating which is the correct picture. In this task, a target figure and five alternatives below were shown, and the participant had to select among the five alternatives, which are quite similar to the target, the one that is identical to the target. The task involves five alternatives and is comprised of two practice items and 20 experimental items; for each item, the number of errors (the number of times in which the participant pointed at a wrong picture) was recorded (MFFT errors, expected range 0–100); in addition, the RT for the first response in each trial was recorded (MFFT RT). The test–retest reliability reported for this task in a sample of primary school children was .49 [53].

*Flanker task.* This task is considered a measure of interference control. We used a computerized version of the task where stimuli and procedures were similar to those used in previous studies [54, 55]. Participants completed a series of items in which they were shown five fish in one horizontal row and are instructed to pay attention to the fish in the middle, which is the target fish. On each trial, they were asked to respond as quickly as possible as to whether the target fish was looking to the left or right, pressing the left or the right bottom respectively. The other four fish flanking the middle fish (the flankers) can either look in the same

(compatible) or the opposite direction (incompatible) as the target fish. The target fish was always located in the same location (i.e., the centre of the display) on every trial. For each trial, stimuli were presented until a response was made or until more than 3000 ms elapsed. Each trial was presented after 1500 ms. After an intertrial interval of 1500 ms, a new trial was presented. If a participant responded in less than 200 ms, this was considered an anticipatory error in line with Christ [54]. Children completed two practice blocks of 20 items each (in the first practice block, the target fish was presented alone, while in the second practice block the target fish and the four flanking fish were shown). After these two practice blocks, children completed a total of 120 experimental items (60 compatible and 60 incompatible items) that were randomly intermixed. At intervals of 40 items, children were offered a one-minute break. The proportion of correct responses and the response times in the incongruent and congruent items were recorded. Regarding response times, we only considered RT in the correct items (Flanker accuracy, expected range 0–1; Flanker RT, expected range-200-3000 ms). Split-half reliability ranged from 0.34 to 0.42 was reported in a sample of adolescents [56].

**Mathematical measures.** To assess specific math abilities, we used four types of tasks (arithmetic facts, mental calculation, inferences and math problem solving) taken from Italian standardized mathematical batteries chosen according to the age of participants. Arithmetic facts and mental calculation can be administered to the entire age range of our sample, whereas inferences and math problem solving were not available for participants of high school and university. As the number of item and the level of difficulty varied with age, z scores were used for all math tasks.

**Arithmetic facts (AC-MT 3 [57] for primary and middle school students; MT-3 Advanced Clinical, [58] for high school students; LSC-SUA [59] for participants over 19 years).** This test assesses the ability to memorize and retrieve arithmetic facts (i.e., whether the participant already has the information available in memory and can access it without performing calculation procedures). Participants are orally presented with simple operations, to which they must respond as quickly as possible, within three seconds. Each item can only be repeated once. The number of items and the level of difficulty vary based on the corresponding school grade. Test-retest reliability (Pearson's r) calculated in 215 TD primary and middle school students was .91. The total score was the number of correct items within time limit, transformed into z scores according to test norms for each school grade.

**Mental calculation (AC-MT 3 [57] for primary and middle school students; MT-3 Advanced Clinical [58] for high school students; LSC-SUA [59] for participants over 19 years).** This task assesses the child's ability to apply mental calculation strategies to arrive at the correct result of an operation. Participants are required to mentally solve operations presented orally, within 30 seconds. Each item can only be repeated once. Test-retest reliability (Pearson's r) calculated in 211 TD primary and middle school students was .80 The total score was the number of correct items within time limit, transformed into z scores according to the standardized norms for each school grade.

**Inferences (AC-MT 3 [57]).** This test investigates the partic=ipant's ability to perform inferential mathematical reasoning, their understanding of mathematical symbols, and the degree of automation of arithmetic procedures and fundamental principles. The subtest is divided into three different tasks. The first one required to solve calculation with figures (e.g., in the operation "flower + flower = 8", the participant has to understand that one flower is equivalent to 4). In the second task, the participant had to add the missing mathematical symbol in numerical operations. In the last, the third task, there are two operations: one complete, while in the other the result is missing. Students were required to complete the calculation using the second operation as an aid. The total time available for the subscale was two minutes, one minute for the first type of the task and one minute for the other two tasks. Test-retest

reliability (Pearson's r) calculated in 198 TD primary and middle school students was .69. The total score was the number of correct items within time limit, transformed into z scores according to the standardized norms for each school grade.

**Math problem solving (AC-MT 6–11, [60]; AC-MT 11–14 [61]).** In this task participants had to solve arithmetic word problems, presented in a written form. During the task, they were able to write the resolution on a paper. Five math problems were administered to primary students (from 3rd to 5th grade) and ten math problems were administered to middle school students; the time limit for this task was 40 minutes for primary school and 30 minutes for middle school. Cronbach's alphas between .68 to.73 were reported in samples of primary school students. For each solution, one point was given if both the procedure and calculation were correct, and 0.5 point was given if only the procedure was correct. Z scores calculated according to the standardized norms for each school grade were used.

**Analytic strategy.** All the analyses were performed with Jamovi software, version 2.3.18. Descriptive statistics and zero-order correlations among math measures were computed. To assess groups' differences in math measures (research question a), a t-test for each math ability was conducted to compare the group with ASD and the TD group. Then, we investigated the effect of group (ASD or TD) and domain-general cognitive processes on specific math abilities and examined if the contribution of cognitive processes varied according to the group (research question b). To this end, we first investigated zero-order correlations (Pearson) between cognitive processes and specific math abilities. Math abilities variables were expressed in z scores, as different items were administered to different school age. To take the participants different ages into account, we calculated residual scores for each cognitive process running a series of regression analysis with age as predictor and raw score of each cognitive process as dependent variable [62, 63]. To determine the contribution of group and each cognitive process to specific math abilities, a series of hierarchical linear regression analyses were conducted; each math ability was used as dependent variable, whereas the independent variables were included in two three blocks: group (ASD or TD) and vocabulary in the first block, residual scores of cognitive processes (included one by one in separate regressions) in the second block, while the interaction between group and the cognitive processes was added in the third block. Before running the analysis, we verified that all the necessary assumptions of regression were met, and then, when evaluating the models, we verified there were no collinearity problems (tolerance values were greater than .50, VIF < 2, and condition indices were less than 4.2; Durbin-Watson values ranged from 1.5 to 2.2).

## Results

### Group differences on specific math abilities (research question a)

Table 2 showed the descriptive statistics (N, mean, SD, minimum and maximum) of mathematical measures for the group with ASD and the TD group. The numerosity were lower for inferences and math problem solving as these tasks were administered to participants from the second (inferences) or third grade (math problem solving) of primary school to the third grade of middle school. To assess group differences on each math ability, a t-test was conducted to compare the group with ASD and the TD group. As shown in Table 2, the group with ASD showed lower scores on all the math measures, with a large effect size (Cohen's d).

### Zero order (Pearson) correlations between math abilities and domain-general cognitive processes

Table 3 showed zero-order (Pearson) correlations between specific math abilities and domain-general processes. High correlations among all math measures were found, with values ranging

**Table 2. Specific math abilities of the group with ASD and the TD group.**

| Task | ASD group | | | | | TD group | | | | | t test | Cohen's d |
|---|---|---|---|---|---|---|---|---|---|---|---|---|
| | N | M | SD | Min | Max | N | M | SD | Min | Max | | |
| Arithmetic facts | 24 | -1.63 | 1.22 | -3.50 | 1.09 | 52 | 0.18 | 0.71 | -1.35 | 1.36 | t (74) = 8.12 *** | 2.00 |
| Mental Calculation | 24 | -1.33 | 1.12 | -2.71 | 1.17 | 52 | 0.34 | 0.88 | -1.73 | 1.81 | t (74) = 7.05 *** | 1.74 |
| Inferences | 21 | -1.66 | 1.22 | -2.84 | 1.71 | 34 | 0.17 | 0.88 | -1.91 | 1.55 | t (53) = 6.41*** | 1.78 |
| Math problem solving | 18 | -1.58 | 0.99 | -2.53 | 1.37 | 27 | -0.17 | 0.92 | -1.46 | 1.58 | t (43) = 4.91 *** | 1.49 |

Note: ASD = autism spectrum disorder; TD = Typical Development; M = mean; SD = Standard Deviation

*p < .05

**p < .01

***p < .001.

from 0.58 to 0.85. Most cognitive variables were statistically significant correlated with all considered specific math abilities; in particular, vocabulary, VWM, VSWM, MFFT errors, Flanker AI, were significantly correlated with Arithmetic Facts, Mental Calculation, Mathematical Inferences and Math Problem Solving. No significantly correlations were found between between RT indices of inhibitory measures (MFFT RT, Flanker RTI) and math abilities.

## Contribution of group and cognitive processes on specific math abilities (research question b)

As shown in Table 4, a series of hierarchical linear regression analyses were conducted with each math ability used as dependent variable. For each dependent variable, the following regression were conducted:

- in the first hierarchical linear regression, the contribution of group and vocabulary was investigated, including them as independent variable in the first step and interaction between the two included in the second step;

**Table 3. Zero-order (Pearson) correlations among specific math abilities and domain-general cognitive processes.**

| | Arithmetic Facts | Mental Calculation | Mathematical Inferences | Math Problem Solving |
|---|---|---|---|---|
| Arithmetic Facts | | 0.77*** | 0.85*** | 0.69*** |
| Mental Calculation | | - | 0.66*** | 0.58*** |
| Mathematical Inferences | | | - | 0.78*** |
| Matrix Reasoning | 0.22 | 0.17 | 0.17 | 0.29* |
| Vocabulary | 0.39 *** | 0.27* | 0.36** | 0.54*** |
| BDS | 0.35** | 0.44*** | 0.36** | 0.46** |
| Mr. Peanut | 0.27* | 0.39** | 0.38** | 0.37* |
| MFFT errors | -0.25* | -0.33** | -0.28* | -0.41** |
| MFFT RT | -0.17 | -0.13 | -0.12 | -0.03 |
| Flanker AI | 0.29* | 0.32** | 0.49*** | 0.44** |
| Flanker RTI | -0.05 | -0.13 | -0.08 | -0.01 |

Note: MR = Matrix Reasoning; VC = Vocabulary; BDS = Backward digit span; MFFT errors = number of errors on Matching Familiar Figures Task; MFFT

RT = reaction time on Matching Familiar Figures Task; Flanker AI = Flanker accuracy on incongruent items; Flanker RTI = Flanker reaction time on incongruent items

*p < .05

**p < .01

***p < .001.

**Table 4. Hierarchical linear regression analysis.**

| | | Arithmetic facts | | | Mental Calculation | | | Inferences | | | Math problem solving | | |
|---|---|---|---|---|---|---|---|---|---|---|---|---|---|
| | | F(3,72) = 28.75 p < .001, $R^2_{adj}$ = 0.53 $R^2\Delta$ = 0.03*** | | | F(3,72) = 21.31 p < .001, $R^2_{adj}$ = 0.45 $R^2\Delta$ = 0.06** | | | F(3,51) = 14.60 p < .001 $R^2_{adj}$ = 0.43 $R^2\Delta$ = 0.01 | | | F(3,41) = 11.51 p < .001 $R^2_{adj}$ = 0.42 $R^2\Delta$ < .001 | | |
| | IV | b | SE | β | b | SE | β | B | SE | β | b | SE | B |
| Block 1 | Group | -1.65 | 0.22 | **-1.34***** | -1.60 | 0.25 | **-1.30***** | -1.69 | 0.30 | **-1.24***** | -1.05 | 0.30 | **-0.90**** |
| | Vocabulary | 0.09 | 0.04 | **0.21*** | 0.04 | 0.04 | 0.09 | 0.06 | 0.05 | 0.14 | 0.12 | 0.04 | **0.35**** |
| Block 1&2 | Group | -3.36 | 0.77 | **-1.27***** | -3.92 | 0.84 | **-1.21***** | -2.58 | 1.12 | -1.23* | -0.87 | 1.08 | -0.91 |
| | Vocabulary | 0.02 | 0.04 | 0.05 | -0.05 | 0.05 | -0.13 | 0.02 | 0.07 | 0.05 | 0.13 | 0.07 | 0.37 |
| | Group*Vocabulary | 0.16 | 0.07 | **0.39*** | 0.22 | 0.07 | 0.52** | 0.08 | 0.09 | 0.18 | -0.02 | 0.09 | -0.05 |

| | | Arithmetic facts | | | Mental Calculation | | | Inferences | | | Math problem solving | | |
|---|---|---|---|---|---|---|---|---|---|---|---|---|---|
| | | F(4,71) = 29.39 p = < .001, $R^2_{adj}$ = 0.60 $R^2\Delta$ = 0.09***, 0.03* | | | F(4,71) = 19.51 p = < .001, $R^2_{adj}$ = 0.59 $R^2\Delta$ = 0.11***, 0.01 | | | F(4,50) = 17.31 p = < .001, $R^2_{adj}$ = 0.55 $R^2\Delta$ = 0.09**, 0.04* | | | F(4,40) = 14.10 p = < .001, $R^2_{adj}$ = 0.55 $R^2\Delta$ = 0.10**, 0.03 | | |
| | IV | b | SE | β | b | SE | β | B | SE | β | b | SE | B |
| Block 1 | Group | -1.65 | 0.22 | **-1.34***** | -1.60 | 0.25 | **-1.30***** | -1.69 | 0.30 | **-1.24***** | -1.05 | 0.30 | **-0.90**** |
| | Vocabulary | 0.09 | 0.04 | **0.21*** | 0.04 | 0.04 | 0.09 | 0.06 | 0.05 | 0.14 | 0.12 | 0.04 | **0.35**** |
| Block 1&2 | Group | -1.52 | 0.21 | **-1.24***** | -1.48 | 0.23 | **-1.20***** | -1.55 | 0.28 | **-1.15***** | -0.96 | 0.28 | **-0.82**** |
| | Vocabulary | 0.07 | 0.03 | **0.18*** | 0.02 | 0.04 | 0.06 | 0.06 | 0.04 | 0.13 | 0.11 | 0.04 | **0.30*** |
| | VWM | 0.21 | 0.05 | **0.31***** | 0.22 | 0.06 | **0.32***** | 0.24 | 0.08 | **0.30**** | 0.22 | 0.07 | **0.32**** |
| Block 1&2&3 | Group | -1.58 | 0.20 | **-1.24***** | -1.51 | 0.23 | **-1.20***** | -1.72 | 0.28 | **-1.16***** | -1.12 | 0.28 | **-0.85***** |
| | Vocabulary | 0.06 | 0.03 | 0.15 | 0.02 | 0.04 | 0.04 | 0.03 | 0.04 | 0.08 | 0.09 | 0.04 | **0.26*** |
| | VWM | 0.10 | 0.07 | 0.15 | 0.16 | 0.08 | **0.24*** | 0.08 | 0.10 | 0.11 | 0.07 | 0.11 | 0.10 |
| | Group*VWM | 0.24 | 0.10 | **0.36*** | 0.13 | 0.12 | 0.19 | 0.32 | 0.15 | **0.41*** | 0.27 | 0.14 | 0.39 |

| | | Arithmetic facts | | | Mental Calculation | | | Inferences | | | Math problem solving | | |
|---|---|---|---|---|---|---|---|---|---|---|---|---|---|
| | | F(4,66) = 20.50 p = < .001, $R^2_{adj}$ = 0.54 $R^2\Delta$ = 0.04*, 0.01 | | | F(4,66) = 15.36 p = < .001, $R^2_{adj}$ = 0.45 $R^2\Delta$ = 0.05*, 0.02 | | | F(4,46) = 14.08 p = < .001, $R^2_{adj}$ = 0.51 $R^2\Delta$ = 0.03, 0.01 | | | F(4,36) = 8.98 p = < .001, $R^2_{adj}$ = 0.44 $R^2\Delta$ = 0.01, 0.04 | | |
| | | B | SE | Beta | B | SE | Beta | B | SE | Beta | B | SE | Beta |
| Block 1 | Group | -1.74 | 0.23 | **-1.43***** | -1.69 | 0.26 | **-1.37***** | -1.88 | 0.29 | **-1.41***** | -1.11 | 0.32 | **-0.94**** |
| | Vocabulary | 0.08 | 0.04 | **0.18*** | 0.03 | 0.04 | 0.07 | 0.06 | 0.05 | 0.13 | 0.14 | 0.05 | **0.35*** |
| Block 1&2 | Group | -1.61 | 0.23 | **-1.32***** | -1.53 | 0.26 | **-1.24***** | -1.69 | 0.31 | **-1.27***** | -1.00 | 0.34 | **-0.85**** |
| | Vocabulary | 0.07 | 0.04 | 0.16 | 0.02 | 0.04 | 0.04 | 0.06 | 0.05 | 0.12 | 0.14 | 0.05 | **0.34*** |
| | VSWM | 0.10 | 0.04 | **0.20*** | 0.11 | 0.05 | **0.23*** | 0.11 | 0.07 | 0.17 | 0.06 | 0.07 | 0.12 |
| Block 1&2&3 | Group | -1.53 | 0.23 | **-1.26***** | -1.44 | 0.26 | **-1.17***** | -1.65 | 0.32 | **-1.24***** | -0.97 | 0.33 | **-0.81**** |
| | Vocabulary | 0.07 | 0.04 | 0.15 | 0.01 | 0.04 | 0.03 | 0.06 | 0.05 | 0.13 | 0.14 | 0.05 | **0.36*** |
| | VSWM | 0.06 | 0.05 | 0.13 | 0.07 | 0.05 | 0.14 | 0.05 | 0.10 | 0.07 | -0.06 | 0.10 | -0.11 |
| | Group*VSWM | 0.12 | 0.09 | 0.25 | 0.16 | 0.10 | 0.33 | 0.12 | 0.13 | 0.19 | 0.22 | 0.13 | 0.42 |

| | | Arithmetic facts | | | Mental Calculation | | | Inferences | | | Math problem solving | | |
|---|---|---|---|---|---|---|---|---|---|---|---|---|---|
| | | F(4,68) = 20.43 p = < .001, $R^2_{adj}$ = 0.52 $R^2\Delta$ = < .001, 0.01 | | | (4,68) = 15.48 p = < .001, $R^2_{adj}$ = 0.45 $R^2\Delta$ = 0.01, 0.02 | | | F(4,47) = 14.07 p = < .001, $R^2_{adj}$ = 0.50 $R^2\Delta$ = 0.01, 0.03 | | | F(4,37) = 9.03 p = < .001, $R^2_{adj}$ = 0.44 $R^2\Delta$ = 0.03, 0.01 | | |
| | | B | SE | Beta | B | SE | Beta | B | SE | Beta | B | SE | Beta |
| Block 1 | Group | -1.68 | 0.23 | **-1.36***** | -1.57 | 0.25 | **-1.28***** | -1.79 | 0.30 | **-1.31***** | -1.02 | 0.31 | **-0.86**** |
| | Vocabulary | 0.09 | 0.04 | **0.22*** | 0.06 | 0.04 | 0.15 | 0.07 | 0.05 | 0.16 | 0.14 | 0.05 | **0.37**** |
| Block 1&2 | Group | -1.64 | 0.23 | **-1.32***** | -1.52 | 0.25 | **-1.24***** | -1.73 | 0.30 | **-1.26***** | -0.90 | 0.32 | **-0.76**** |
| | Vocabulary | 0.09 | 0.04 | **0.20*** | 0.05 | 0.04 | 0.12 | 0.06 | 0.05 | 0.13 | 0.12 | 0.05 | **0.33*** |
| | MFFT | -0.02 | 0.02 | -0.07 | -0.02 | 0.02 | -0.08 | -0.03 | 0.03 | -0.10 | -0.05 | 0.03 | -0.20 |
| Block 1&2&3 | Group | -1.66 | 0.23 | **-1.30***** | -1.55 | 0.25 | **-1.20*** | -1.84 | 0.30 | **-1.26***** | -1.01 | 0.34 | **-0.78**** |

*(Continued)*

**Table 4.** (Continued)

| | | Arithmetic facts | | | Mental Calculation | | | Inferences | | | Math problem solving | | |
|---|---|---|---|---|---|---|---|---|---|---|---|---|---|
| | Vocabulary | 0.09 | 0.04 | **0.21***  | 0.06 | 0.04 | 0.14 | 0.06 | 0.05 | 0.14 | 0.12 | 0.05 | **0.33***  |
| | MFFT | 0.02 | 0.03 | 0.08 | 0.05 | 0.04 | 0.19 | 0.03 | 0.04 | 0.14 | -0.01 | 0.05 | -0.05 |
| | Group* MFFT | -0.06 | 0.04 | .25 | -0.11 | 0.05 | -0.25 | -0.10 | 0.05 | 0.38 | -0.05 | 0.07 | -0.21 |
| | | $F_{(4,63)} = 18.50$ $p = < .001$, $R^2_{adj} = 0.52$ $R^2\Delta = 0.01, 0.01$ | | | $F_{(4,63)} = 13.6$ $p = < .001$, $R^2_{adj} = 0.43$ $R^2\Delta = 0.01, 0.01$ | | | $F_{(4,44)} = 17.18$ $p = < .001$, $R^2_{adj} = 0.57$ $R^2\Delta = 0.04^*, 0.01$ | | | $F_{(4,35)} = 10.7$ $p = < .001$, $R^2_{adj} = 0.50$ $R^2\Delta = 0.04, 0.01$ | | |
| | | B | SE | Beta | B | SE | Beta | B | SE | Beta | B | SE | Beta |
| Block 1 | Group | -1.74 | 0.23 | **-1.42*** | -1.76 | 0.26 | **-1.42*** | -1.91 | 0.30 | **-1.39*** | -1.13 | 0.32 | **-0.94*** |
| | Vocabulary | 0.08 | 0.04 | **0.18*** | 0.02 | 0.04 | 0.04 | 0.07 | 0.05 | 0.15 | 0.14 | 0.05 | **0.37**** |
| Block 1&2 | Group | -1.65 | 0.24 | **-1.35*** | -1.67 | 0.27 | **-1.34*** | -1.61 | 0.32 | **-1.17*** | -0.87 | 0.34 | **-0.72*** |
| | Vocabulary | 0.08 | 0.04 | **0.18*** | 0.02 | 0.04 | 0.03 | 0.08 | 0.05 | 0.17 | 0.15 | 0.05 | **0.39**** |
| | Flanker | 1.90 | 1.60 | 0.11 | 1.91 | 1.77 | 0.10 | 3.85 | 1.76 | **0.23*** | 3.26 | 1.77 | 0.23 |
| Block 1&2&3 | Group | -1.65 | 0.24 | **-1.33*** | -1.67 | 0.27 | **-1.33*** | -1.65 | 0.32 | **-1.20*** | -0.92 | 0.34 | **-0.78*** |
| | Vocabulary | 0.07 | 0.04 | **0.16*** | 0.01 | 0.04 | 0.02 | 0.07 | 0.05 | 0.15 | 0.14 | 0.05 | **0.37**** |
| | Flanker | 0.05 | 2.44 | -0.03 | 0.21 | 2.72 | 0.01 | 0.83 | 3.41 | 0.05 | 0.03 | 3.95 | < ..001 |
| | Group* Flanker | 4.20 | 3.25 | 0.23 | 2.96 | 3.62 | 0.16 | 4.10 | 3.98 | 0.25 | 4.04 | 4.41 | 0.29 |

Note: VWM = Verbal Working memory (Backward digit span); VSWM = visuospatial working memory (Mr. Peanut), MFFT = number of errors on Matching Familiar Figures Task; Flanker AI = Flanker accuracy on incongruent items

*$p < .05$

**$p < .01$

***$p < .001$

- in the other hierarchical linear regressions, group and vocabulary (here used as a covariate as the two groups significantly differed on this measure) were included as independent variable in the first step; residual scores of cognitive processes (included one by one in separate regressions) were included in the second step; the interaction between group and the cognitive process was added in the third step. In these analyses, we did not include as predictors the cognitive variables that were not significantly correlated with any math ability.

**Contribution of group and cognitive processes on arithmetic facts.** In the following hierarchical linear regression analyses, arithmetic facts were used as dependent variable.

*Contribution of group and vocabulary to arithmetic facts*. The first regression model, with group and vocabulary as independent variable in the first step, explained 50% of variance and both the variable were significantly predictors. In the second step, the inclusion of interaction between vocabulary and group significantly improved the amount of explained variance; the final model explained 53% of variance and showed that the association between vocabulary and arithmetic facts was stronger in the group with ASD than in the TD group.

*Contribution of group, vocabulary and VWM to arithmetic facts*. VWM, entered as independent variable in the second step, was a significant predictor and significantly improved the amount of explained variance compared to the first step in which only group and vocabulary were included. The interaction between group and VWM in the third block was also a significant predictor and improved the amount of explained variance. The final model explained 60% of variance and showed that the association between VWM and arithmetic facts was stronger in the group with ASD than in the TD group.

*Contribution of group, vocabulary and VSWM to arithmetic facts*. Also VSWM, entered as independent variable in the second step, was a significant predictor and significantly improved

the amount of explained variance, compared to the first step in which only group and vocabulary were included. The interaction between group and VSWM in the third block was not a significant predictor and did not improve the amount of explained variance. The final model explained 54% of variance and showed that the association between VSWM and arithmetic facts was similar in the two groups.

*Contribution of group, vocabulary and inhibitory measures to arithmetic facts.* Either the inhibitory measures, i.e., MFFT errors and Flanker AI, and their interaction with group were not significantly predictors of arithmetic facts and did not significantly improve the amount of explained variance.

**Contribution of group and cognitive processes on mental calculation.** In the following hierarchical linear regression analyses, mental calculation was used as dependent variable.

*Contribution of group and vocabulary to mental calculation.* The first regression model, with group and vocabulary as independent variable in the first step, explained 39% of variance and only group, but not vocabulary, was a significant predictors; in the second step, interaction between vocabulary and group was a significant predictor and improved the amount of explained variance. The final model explained 45% of variance and showed that the association between vocabulary and mental calculation was stronger in the group with ASD than in the TD group.

*Contribution of group, vocabulary and VWM to mental calculation.* VWM, entered as independent variable in the second step, was a significant predictor and significantly improved the amount of explained variance compared to the first step in which only group and vocabulary were included. The interaction between group and VWM in the third block was not a significant predictor and did not improve the amount of explained variance. The final model explained 59% of variance and indicated that the association between VWM and mental calculation was similar in the two groups.

*Contribution of group, vocabulary and VSWM to mental calculation.* VSWM, entered as independent variable in the second step, was a significant predictor; the model explained 43% of variance and significantly improved the amount of explained variance, compared to the first step in which only group and vocabulary were included. The interaction between group and VSWM in the third block was not a significant predictor and did not improve the amount of explained variance. The final model explained 45% of variance and showed that the association between VSWM and mental calculation was similar in the two groups.

*Contribution of group, vocabulary and inhibitory measures to mental calculation.* Either the inhibitory measures, i.e., MFFT errors and Flanker AI, and their interaction with group, were not significantly predictors of mental calculation and did not significantly improve the amount of explained variance.

**Contribution of group and cognitive processes on inferences.** In the following hierarchical linear regression analyses, inferences were used as dependent variable.

*Contribution of group and vocabulary to inferences.* The first regression model, with group and vocabulary as independent variable in the first step, explained 42% of variance and only group but not vocabulary was a significant predictor. In the second step, interaction between vocabulary and group was not a significant predictor and did not improve the amount of explained variance. The final model explained 43% of variance.

*Contribution of group, vocabulary and VWM to inferences.* VWM, entered as independent variable in the second step, was a significant and significantly improved the amount of explained variance in addition to the first step in which only group and vocabulary were included. In addition, the interaction between group and VWM in the third block was also a significant predictor and significantly improved the amount of explained variance. The final model explained 55% of variance and showed that the association between VWM and inferences was stronger in the group with ASD than in the TD group.

*Contribution of group, vocabulary and VSWM to inferences.* VSWM, entered as independent variable in the second step, was not a significant predictor and did not improve the amount of explained variance compared to the first step in which only group and vocabulary were included. The interaction between group and VSWM in the third block was not a significant predictor and did not improve the explained variance. The final model explained 51% of variance.

*Contribution of group, vocabulary and inhibitory measures to inferences.* Among inhibitory measures, Flanker AI, included in the second step, was a significant predictor and improve the amount of explained variance compared to the first step in which only group and vocabulary were included. The interaction between group and Flanker AI in the third block was not significant and did not improve the explained variance suggesting that the association between Flanker and inferences was similar in the two groups. The final model explained 57% of variance.

The other inhibitory measure (MFFT errors) and its interaction with group was not significant predictor of inferences and did not significantly improve the amount of explained variance.

**Contribution of group and cognitive processes on math problem solving.** In the following hierarchical linear regression analyses, math problem solving was used as dependent variable.

*Contribution of group and vocabulary to math problem solving.* The first regression model, with group and vocabulary as independent variable in the first step, explained 41% of variance and both group and vocabulary were significant predictors. In the second step, interaction between vocabulary and group was not a significant predictor and did not improve the amount of explained variance, showing that the association between vocabulary and math problem solving was similar in the two groups. The final model explained 42% of variance.

*Contribution of group, vocabulary and VWM to math problem solving.* VWM, entered as independent variable in the second step, was a significant predictor and significantly improved the amount of explained variance compared to the first step in which only group and vocabulary were included. The interaction between group and VWM in the third block was not significant and did not improve the amount of explained variance and showed that the association between VWM and math problem solving was similar in the two groups. The final model explained 55% of variance.

*Contribution of group, vocabulary and VSWM to math problem solving.* VSWM, entered as independent variable in the second step, was not a significant predictor and did not improve the amount of explained variance in addition to the first step in which only group and vocabulary were included. The interaction between group and VSWM in the third block was not a significant predictor and did not improve the explained variance. The final model explained 44% of variance.

*Contribution of group, vocabulary and inhibitory measures to math problem solving.* Either the two inhibitory measures, i.e., MFFT (errors) and Flanker AI, and their interaction with group were not significantly predictors of math problem solving and did not significantly improve the amount of explained variance.

In summary, vocabulary was significantly associated to arithmetic facts, mental calculation and problem solving, but not to inferences. In both arithmetic facts and mental calculation, this association was stronger in the group with ASD. VWM was significantly associated to all the specific math abilities, and in arithmetic facts and inferences this association was stronger in the group with ASD. VSWM was associated with arithmetic facts and mental calculation and its contribution was similar in the two groups. Among inhibitory measures, Flanker AI was associated with inferences and its contribution was similar in the two groups.

## Discussion

The current study aimed to examine specific mathematical abilities (research question a) and how they be influenced by domain-general cognitive processes (research question b) in a group with ASD and a TD group. The results demonstrated that the group with ASD exhibited lower performance across all mathematical tasks considered, with a comparable large effect. This supports the conclusions of a previous meta-analysis, which found no significant differences based on the type of math task [23]. In this study, it's important to note that the task type includes not just numerical operations and math problem-solving, but also encompasses arithmetic facts and the capacity to draw mathematical inferences.

Beyond uncovering a generalized deficit across various mathematical abilities, our results offer valuable insights into understanding the processes that contribute to these abilities in both typical development and autism. Overall, our findings closely align with the model proposed by Cragg and coauthors [6]. In fact, our study identified a crucial role of verbal working memory, a significant predictor for all specific math tasks. Additionally, we examined vocabulary, which proved to be a significant predictor for all evaluated mathematical abilities, except for inferences.

Consistent with Cragg et al.'s [6] research, our study found that both verbal and visuospatial working memory play roles in arithmetic facts and calculations. Specifically, verbal working memory was linked to factual knowledge, while visuospatial working memory was associated with procedural knowledge. On the other hand, only verbal working memory was associated with tasks requiring more conceptual knowledge, like inferences and mathematical problem-solving. This finding supports the notion that working memory is crucial for accessing information stored in long-term memory, suggesting that conceptual information is predominantly stored in a verbal format [6]. Students with low verbal working memory capacity are less likely to choose a retrieval strategy in math problem solving tasks and are also likely to retrieve them less accurately [64, 65]. Arithmetic facts, though primarily conceptualized in terms of numerical information, may also incorporate a visuospatial component. This connection is likely influenced by factors such as the format of presentation or the utilization of visual aids (e.g., using charts to memorize times tables) during the encoding process [6, 66].

Notably, the current study showed that these relationships may have a different strength in the group with ASD and the TD group. In fact, in arithmetic facts and inferences the effect of verbal working memory was stronger in the group with ASD, whereas in mental calculation and math problem solving the contribution was similar in the two groups. A quite similar result was found considering vocabulary, as its contribution was stronger for the group with ASD in arithmetic fact and mental calculation, whereas was similar in the two groups in math problem solving. The fundamental role of vocabulary and verbal working memory confirmed the results from the previous meta-analysis [23] and previous studies with participants with ASD [20, 22, 35]. It is well known that language and working memory facilitates knowledge retrieval from long-term memory during mathematics performance [67]. However, in typical development, as students cumulatively build their mathematics knowledge, the direct retrieval of arithmetic facts from long-term memory can reduce cognitive load and, therefore, working memory demands [68]. Thus, when the retrieval of arithmetic facts becomes more automatic, verbal ability and working memory become less relevant. However, the current study seemed to suggest a different pattern for students with ASD; their performance on arithmetic facts was predicted by both verbal ability and verbal working memory, with a stronger association than in the TD group [67, 68]. Therefore, for students with ASD, arithmetic facts appeared to pose greater challenges and were less automatized compared to their peers. The retrieval of these facts seemed to depend more on verbal resources. Instead, the contribution of both vocabulary

and verbal working memory to math problem solving was similar in the two groups. This result was in line with previous studies suggesting that the direct retrieval of general knowledge stored in long-term memory can enhance the understanding of word problems [69, 70]. The results showed that solving math problems was generally a complex ability that place high demands on working memory for both students with ASD and their TD peers.

We also examined how inhibitory control was related to specific math skills. Differently from Cragg's model [6], we didn't find inhibitory control contributing to arithmetic facts or mental calculation. However, it's worth noting that in Cragg's study, only a response inhibition task with numerical stimuli was a significant predictor, while the response inhibition task without numerical stimuli wasn't linked to any math variables. In our study, inhibitory tasks didn't involve numerical stimuli because it's known that these are more closely related to math tasks [7, 32]. Additionally, unlike most previous studies, we used not only response inhibition measures but also interference control measures. This allowed us to see how the ability to filter out irrelevant stimuli supports more conceptually based math skills such as mathematical inferences. Thus, it's possible that interference control, rather than just response inhibition, might play a significant role in certain areas of math learning [32]. Moreover, the results suggest that the association between interference control (in our case, accuracy on incongruent Flanker items) and mathematical inferences is statically significant and similar in both the group with ASD and the TD group.

## Limitations and future directions

The study's findings should be considered in light of some limitations. First, the limited sample size of the group with ASD reduces statistical power and the wide age range prevents control over confounding variables, such as varying treatment types and durations based on participant cohorts. Regarding the procedure, online administration posed challenges due to participants' lack of appropriate technology. To ensure controlled assessment, all tasks, including computerized ones, were conducted via screen sharing. Lastly, there are limitations related to the tests used. Using only two tasks (Reason Matrix and Vocabulary) to investigate intellectual domains is a limitation due to session length. We selected the most representative tasks from these domains to avoid participants' overwhelming. Also concerning working memory measures, it's important to note that verbal working memory task was a digit span task and therefore the use of numerical stimuli could have increased the association with math tasks. In addition, a larger sample size would have also allowed the investigation of the contribution of each domain-general processes, while controlling for the effect of the others.

The findings of the present study also suggested practical implications. To provide effective education that enhances academic achievement in students with ASD, a deeper understanding of factors influencing individual differences in academic performance is required, considering both domain-general and domain-specific processes [26, 38, 71]. Additionally, these students benefit from general education, but they may need multidimensional assessments to pinpoint strengths, weaknesses and additional support to improve academic outcomes of students with ASD [71]. Moreover, it is crucial to translate research findings into intervention strategies. On one hand, it is essential to adopt compensatory strategies such as reducing linguistic working memory demands, facilitating task focus, creating suitable settings with the use of visual aids to support students in remembering procedures and focussing on relevant aspects. On the other hand, it's important to implement interventions that enhance both domain-general and domain-specific processes and focus on the automatization of procedures and strategies that can reduce the cognitive load, especially for those students with difficulties in WM.

## Conclusions

In conclusion, the current study suggested the importance of considering both domain-specific and domain-general aspects from a multidimensional perspective, even in the context of atypical development. Advancing toward integrating domain-general processes and domain-specific abilities is crucial in research involving atypical development to better understand these complex skills and the reasons behind the challenges faced by students with ASD.

## Supporting information

**S1 Data.**
(XLSX)

## Author Contributions

**Conceptualization:** Irene Tonizzi, M. Carmen Usai.

**Data curation:** Irene Tonizzi.

**Project administration:** M. Carmen Usai.

**Supervision:** M. Carmen Usai.

**Visualization:** Irene Tonizzi.

**Writing – original draft:** Irene Tonizzi.

**Writing – review & editing:** Irene Tonizzi, M. Carmen Usai.

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
