## [Decision Letter · Decision Letter 0]

23 Jun 2024

PONE-D-24-09141Cognitive correlates of math abilities in autism spectrum disorderPLOS ONE

Dear Dr. Tonizzi,

Thank you for submitting your manuscript to PLOS ONE. After careful consideration, we feel that it has merit but does not fully meet PLOS ONE’s publication criteria as it currently stands. Therefore, we invite you to submit a revised version of the manuscript that addresses the points raised during the review process.

We look forward to receiving your revised manuscript.

Kind regards,

Laura Mandolesi

Academic Editor

PLOS ONE

Reviewers' comments:

Reviewer's Responses to Questions

**Comments to the Author**

1. Is the manuscript technically sound, and do the data support the conclusions?

Reviewer #1: Yes

Reviewer #2: Yes

2. Has the statistical analysis been performed appropriately and rigorously? 

Reviewer #1: Yes

Reviewer #2: Yes

3. Have the authors made all data underlying the findings in their manuscript fully available?

Reviewer #1: Yes

Reviewer #2: Yes

4. Is the manuscript presented in an intelligible fashion and written in standard English?

Reviewer #1: Yes

Reviewer #2: Yes

5. Review Comments to the Author

Reviewer #1: First of all, I appreciate the opportunity to review the article entitled “Cognitive correlates of math abilities in autism spectrum disorder”. It's a commendable piece of work, demonstrating a meticulous and thorough methodological approach. The study's commitment to ensuring the reliability and validity of its findings is evident. Below are just some suggestions to further enhance its impact.

The introduction provides a comprehensive overview of previous literature on mathematics abilities in children with Autism Spectrum Disorder (ASD), including references to relevant studies and meta-analyses. It clearly outlines the objectives of the study, emphasizing the importance of understanding the strengths and weaknesses in mathematical learning in children with ASD and investigating the contributions of inhibitory control (IC) and working memory (WM) to these abilities. By addressing the gaps in existing research, it underscores the need for further investigation in this area.

On the other hand it presents dense information, potentially overwhelming readers and hindering clarity. Simplifying the language and condensing content could improve readability. For example, repetitive discussions on ASD heterogeneity and research limitations are evident, requiring streamlining to enhance coherence. While effectively summarizing existing knowledge, I suggest that the introduction could be strengthened by further emphasizing the contribution to knowledge of the proposed study. Explicit insights into how to fill existing gaps would strengthen its significance.

Regarding the study description, the author clearly outlines the two main aims of the study, providing context on the current state of research and the challenges encountered in analyzing mathematical abilities in children with Autism Spectrum Disorder (ASD). The methodological approach is well defined, intending to examine differences in mathematical abilities between individuals with ASD and typically developing ones through a series of specific mathematical tasks. The methods employed are robust and reflect a rigorous and accurate scientific approach to research.

However, some limitations should be highlighted, such as the representativeness of the samples and the generalizability of the results to larger populations, especially concerning the implications that these findings can have on the individualized interventions that should derive from them. I suggest delving deeper into this aspect (in the discussion or limitation section), citing some studies that highlight the difficulty of finding the most effective appropriate treatment considering the extreme phenotypic variability of autism, especially with reference to the practical and treatment implications, also in other areas of intervention, for example:

Simeoli, R., Cerasuolo, M., Nappo, R., Gallucci, M., Iovino, L., Frolli, A., & Rega, A. (2022). Examining predictors of different ABA treatments: A systematic review. Behavioral Sciences, 12(2), 267.;

Klinger, L. G., Cook, M. L., & Dudley, K. M. (2021). Predictors and Moderators of Treatment Efficacy in Children and Adolescents with Autism Spectrum Disorder. Journal of clinical child and adolescent psychology : the official journal for the Society of Clinical Child and Adolescent Psychology, American Psychological Association, Division 53, 50(4), 517–524. https://doi.org/10.1080/15374416.2020.1833735

Results and general findings are well-supported by existing literature. The study extends previous literature by highlighting the differential contributions of verbal and visuospatial working memory to various math tasks, suggesting a nuanced relationship between cognitive processes and mathematical abilities.

Overall, the manuscript provides valuable insights into the cognitive correlates of math abilities in ASD, shedding light on potential intervention targets and informing educational practices for individuals with neurodevelopmental disorders. The thorough methodology and robust statistical analyses strengthen the study's contribution to the field of cognitive psychology and autism research. However, further discussion on the implications of these findings for educational interventions and future research directions would enhance the manuscript's impact and relevance.

Reviewer #2: The article addresses the topic of math abilities in autism spectrum disorder (ASD), particularly focusing on the cognitive processes that can contribute to ASD math outcomes. This is an important topic, especially given the ongoing interest in individualized academic programs for students with neurodevelopmental disorders. However, several issues with the methods and analysis need to be addressed. Below are more specific comments by section:

Abstract

I recommend adding a short sentence regarding the theoretical framework that supports the study methods.

Introduction

The research question (a) described from lines 206 to 208 should be more detailed, explaining better what kind of differences between the ASD and TD groups the authors aimed to analyze. Besides research questions, the authors have not presented the hypothesis of the current study. It is difficult for the reader to understand the scope of the study.

Method

Regarding the method section, first, I point out that ASD participants’ diagnosis is based on DSM, ICD, SRS, and CARS, which are not gold standard measures for the diagnosis of ASD. This should be a limitation underlined in “Limitations” section.

Additionally, I recommend adding a power analysis to explain how the authors determined the sample size needed for analysis.

Third, from lines 238-242, I suggest briefly describe in narrative form (in the text) how the groups differ on the cognitive tasks mentioned.

Lastly, the method section is missing a crucial procedure: a description of how informed consent was obtained from participants and their parents, information on the ethics committee that approved the study, and a reference to the Declaration of Helsinki. I highly recommend including these details.

Analytic Strategy

From lines 377 to 380, it is reported: “To take the participants' different ages into account, we calculated residual scores for each cognitive process by running a series of regression analyses with age as the predictor and the raw score of each cognitive process as the dependent variable [62, 63].” However, it seems that these analyses are not reported in the results section.

Results

Regarding the “Zero Order (Pearson) correlations between math abilities and domain-general cognitive processes,” I recommend adding multiple corrections (e.g., Bonferroni or Holm-Bonferroni correction).

Discussions

Discussions of the study results seem complete and adequate.

General suggestions

I advise the authors to review the writing to correct some typos and to improve the flow and readability of the text.

Finally, I would ask the authors if they considered possible differences between different levels of severity of ASD (based on DSM criteria) in math abilities and the cognitive processes related to math abilities. This could be a research question for future studies.

6. PLOS authors have the option to publish the peer review history of their article (what does this mean?). If published, this will include your full peer review and any attached files.

Reviewer #1: No

Reviewer #2: **Yes: **Federica Somma

---

## [Author Response · Author response to Decision Letter 0]

18 Aug 2024

PONE-D-24-09141

Cognitive correlates of math abilities in autism spectrum disorder

PLOS ONE

Dear Dr. Tonizzi,

Thank you for submitting your manuscript to PLOS ONE. After careful consideration, we feel that it has merit but does not fully meet PLOS ONE’s publication criteria as it currently stands. Therefore, we invite you to submit a revised version of the manuscript that addresses the points raised during the review process.

We look forward to receiving your revised manuscript.

Kind regards,

Laura Mandolesi

Academic Editor

PLOS ONE

We would like to thank very much the Editor and the Reviewers for the opportunity to revise the manuscript. We are particularly grateful for this opportunity, which allowed us to improve the quality of our paper. We reported our responses below each comment. All changes were highlighted in the revised version of the manuscript.

Reviewers'comments:

Reviewer's Responses to Questions

Comments to the Author

1. Is the manuscript technically sound, and do the data support the conclusions?

Reviewer #1: Yes

Reviewer #2: Yes

2. Has the statistical analysis been performed appropriately and rigorously?

Reviewer #1: Yes

Reviewer #2: Yes

3. Have the authors made all data underlying the findings in their manuscript fully available?

Reviewer #1: Yes

Reviewer #2: Yes

4. Is the manuscript presented in an intelligible fashion and written in standard English?

Reviewer #1: Yes

Reviewer #2: Yes

5. Review Comments to the Author

Please use the space provided to explain your answers to the questions above. You may also include additional comments for the author, including concerns about dual publication, research ethics, or publication ethics. (Please upload your review as an attachment if it exceeds 20,000 characters.

Reviewer #1: First of all, I appreciate the opportunity to review the article entitled “Cognitive correlates of math abilities in autism spectrum disorder”. It's a commendable piece of work, demonstrating a meticulous and thorough methodological approach. The study's commitment to ensuring the reliability and validity of its findings is evident. Below are just some suggestions to further enhance its impact.

The introduction provides a comprehensive overview of previous literature on mathematics abilities in children with Autism Spectrum Disorder (ASD), including references to relevant studies and meta-analyses. It clearly outlines the objectives of the study, emphasizing the importance of understanding the strengths and weaknesses in mathematical learning in children with ASD and investigating the contributions of inhibitory control (IC) and working memory (WM) to these abilities. By addressing the gaps in existing research, it underscores the need for further investigation in this area.

On the other hand it presents dense information, potentially overwhelming readers and hindering clarity. Simplifying the language and condensing content could improve readability. For example, repetitive discussions on ASD heterogeneity and research limitations are evident, requiring streamlining to enhance coherence. While effectively summarizing existing knowledge, I suggest that the introduction could be strengthened by further emphasizing the contribution to knowledge of the proposed study. Explicit insights into how to fill existing gaps would strengthen its significance.

We would like to thank the reviewer for this observation. We have revised the text to make it clearer and more concise, eliminating redundant parts and highlighting the gaps that the study aims to fill

Regarding the study description, the author clearly outlines the two main aims of the study, providing context on the current state of research and the challenges encountered in analyzing mathematical abilities in children with Autism Spectrum Disorder (ASD). The methodological approach is well defined, intending to examine differences in mathematical abilities between individuals with ASD and typically developing ones through a series of specific mathematical tasks. The methods employed are robust and reflect a rigorous and accurate scientific approach to research.

However, some limitations should be highlighted, such as the representativeness of the samples and the generalizability of the results to larger populations, especially concerning the implications that these findings can have on the individualized interventions that should derive from them. I suggest delving deeper into this aspect (in the discussion or limitation section), citing some studies that highlight the difficulty of finding the most effective appropriate treatment considering the extreme phenotypic variability of autism, especially with reference to the practical and treatment implications, also in other areas of intervention, for example:

Simeoli, R., Cerasuolo, M., Nappo, R., Gallucci, M., Iovino, L., Frolli, A., & Rega, A. (2022). Examining predictors of different ABA treatments: A systematic review. Behavioral Sciences, 12(2), 267.;

Klinger, L. G., Cook, M. L., & Dudley, K. M. (2021). Predictors and Moderators of Treatment Efficacy in Children and Adolescents with Autism Spectrum Disorder. Journal of clinical child and adolescent psychology : the official journal for the Society of Clinical Child and Adolescent Psychology, American Psychological Association, Division 53, 50(4), 517–524. https://doi.org/10.1080/15374416.2020.1833735

Thank you for the valuable suggestions. We have delved into the recommended literature and expanded the section on the limitations and implications of the study, highlighting the extreme phenotypic variability of autism and the consequent difficulty in designing individualized interventions.

Results and general findings are well-supported by existing literature. The study extends previous literature by highlighting the differential contributions of verbal and visuospatial working memory to various math tasks, suggesting a nuanced relationship between cognitive processes and mathematical abilities.

Overall, the manuscript provides valuable insights into the cognitive correlates of math abilities in ASD, shedding light on potential intervention targets and informing educational practices for individuals with neurodevelopmental disorders. The thorough methodology and robust statistical analyses strengthen the study's contribution to the field of cognitive psychology and autism research. However, further discussion on the implications of these findings for educational interventions and future research directions would enhance the manuscript's impact and relevance

Thank you, we have expanded the section on implications

Reviewer #2: The article addresses the topic of math abilities in autism spectrum disorder (ASD), particularly focusing on the cognitive processes that can contribute to ASD math outcomes. This is an important topic, especially given the ongoing interest in individualized academic programs for students with neurodevelopmental disorders. However, several issues with the methods and analysis need to be addressed. Below are more specific comments by section:

Abstract

I recommend adding a short sentence regarding the theoretical framework that supports the study methods.

Thank you for the observation. We have included the reference to the multidimensional model in the abstract

Introduction

The research question (a) described from lines 206 to 208 should be more detailed, explaining better what kind of differences between the ASD and TD groups the authors aimed to analyze. Besides research questions, the authors have not presented the hypothesis of the current study. It is difficult for the reader to understand the scope of the study.

We fully agree with the reviewer and have accordingly revised the section on research questions. Specifically, we have clarified them further and added considerations regarding the hypotheses

Method

Regarding the method section, first, I point out that ASD participants’ diagnosis is based on DSM, ICD, SRS, and CARS, which are not gold standard measures for the diagnosis of ASD. This should be a limitation underlined in “Limitations” section.

Thank you. In the limitations section, we have highlighted the importance of using standardized measures for autism, such as ADOS and ADI, in future studies

Additionally, I recommend adding a power analysis to explain how the authors determined the sample size needed for analysis.

Thank you for the suggestion. We have highlighted in the limitations section the need to replicate the study with a larger sample, especially for the group with ASD. In addition, we report here the information regarding the power analysis conducted with jPower before starting the study: considering the average effect size of 0.50 identified in the previous meta-analysis, to achieve a power of at least 0.80, 51 participants per group would have been necessary. This number was reached only in the typical development group, but not in the group with ASD. However, the size of the group with ASD could be considered adequate if we consider the maximum effect size identified by the previous meta-analysis (0.77), which would require 22 participants per group to achieve a power of 0.80.

Third, from lines 238-242, I suggest briefly describe in narrative form (in the text) how the groups differ on the cognitive tasks mentioned.

We have revised the text to briefly describe the differences between the two groups in cognitive tasks.

Lastly, the method section is missing a crucial procedure: a description of how informed consent was obtained from participants and their parents, information on the ethics committee that approved the study, and a reference to the Declaration of Helsinki. I highly recommend including these details.

Thank you for the suggestion. We have included the section on informed consent and the reference to the Declaration of Helsinki

Analytic Strategy

From lines 377 to 380, it is reported: “To take the participants' different ages into account, we calculated residual scores for each cognitive process by running a series of regression analyses with age as the predictor and the raw score of each cognitive process as the dependent variable [62, 63].” However, it seems that these analyses are not reported in the results section.

We have included these analyses in the supplementary materials and added the reference in the text (Table S1)

Results

Regarding the “Zero Order (Pearson) correlations between math abilities and domain-general cognitive processes,” I recommend adding multiple corrections (e.g., Bonferroni or Holm-Bonferroni correction).

Thank you for the comment. We have included the Bonferroni correction in the correlation table

Discussions

Discussions of the study results seem complete and adequate.

General suggestions

I advise the authors to review the writing to correct some typos and to improve the flow and readability of the text.

Finally, I would ask the authors if they considered possible differences between different levels of severity of ASD (based on DSM criteria) in math abilities and the cognitive processes related to math abilities. This could be a research question for future studies.

This is definitely an interesting observation that merits further investigation. Since it would require a larger sample size, we have included it in the limitations section.

6. PLOS authors have the option to publish the peer review history of their article (what does this mean?). If published, this will include your full peer review and any attached files.

Do you want your identity to be public for this peer review? For information about this choice, including consent withdrawal, please see our Privacy Policy.

Reviewer #1: No

Reviewer #2: Yes: Federica Somma

---

## [Editor Report · Decision Letter 1]

3 Sep 2024

Cognitive correlates of math abilities in autism spectrum disorder

PONE-D-24-09141R1

Dear Dr. Tonizzi

We’re pleased to inform you that your manuscript has been judged scientifically suitable for publication and will be formally accepted for publication once it meets all outstanding technical requirements.

Kind regards,

Laura Mandolesi

Academic Editor

PLOS ONE
---

## [Editor Report · Acceptance letter]

6 Sep 2024

PONE-D-24-09141R1 

PLOS ONE

Dear Dr. Tonizzi, 

I'm pleased to inform you that your manuscript has been deemed suitable for publication in PLOS ONE. Congratulations! Your manuscript is now being handed over to our production team.

Kind regards, 

on behalf of

Professor Laura Mandolesi 

Academic Editor

PLOS ONE